Sterilization of image steganography using self-supervised convolutional neural network

Liu Jinjin 1
Xu Fuyong 2
Zhao Yingao 2
Xin Xianwei 2 3
Liu Keren 2
Ma Yuanyuan 121100@htu.edu.cn 2 3
1 Software College of Software, Henan Normal University , Xinxiang , Henan , China
2 College of Computer and Information Engineering, Henan Normal University , Xinxiang , Henan , China
3 Engineering Lab of Intelligence Business & Internet of Things, Henan Normal University , Xinxiang , Henan , China
Coelho Paulo Jorge
Electronic publication date: 2024 Sep 24
Publication date: 2024
Volume: 10
Electronic Location ID: e2330
Received 2024 Feb 2; Accepted 2024 Aug 22
Copyright: ©2024 Liu et al.
Copyright year: 2024
Copyright holder: Liu et al.
License: This is an open access article distributed under the terms of the Creative Commons Attribution License, which permits unrestricted use, distribution, reproduction and adaptation in any medium and for any purpose provided that it is properly attributed. For attribution, the original author(s), title, publication source (PeerJ Computer Science) and either DOI or URL of the article must be cited.
License URL: https://creativecommons.org/licenses/by/4.0/

Keywords: Image steganalysis, Self-supervised learning, Steganography sterilization, Steganography

Funding: The Henan Province Science Foundation for Youths No. 222300420058 The National Natural Science Foundation of China No. 62002103 The Key Research Project for Higher Education Institutions in Henan Province No. 24A520019 This work was supported by the Henan Province Science Foundation for Youths (No. 222300420058), the National Natural Science Foundation of China (No. 62002103) and the Key Research Project for Higher Education Institutions in Henan Province (No. 24A520019). The funders had no role in study design, data collection and analysis, decision to publish, or preparation of the manuscript.

==============================
Background

With the development of steganography technology, lawbreakers can implement covert communication in social networks more easily, exacerbating network security risks. Sterilization of image steganography methods can eliminate secret messages to block the transmission of illegal covert communication. However, existing methods overly rely on cover-stego image pairs and are unable to sanitize unknown image, which reduces stego image blocking rate in social networks.

Methods

To address the above problems, this paper proposes an effective sterilization of image steganography method using self-supervised convolutional neural network (SS-Net), which does not require any prior knowledge of image steganography schemes. SS-Net includes a purification module and a refinement module. Firstly, the pixel-shuffle down-sampling in purification module is adopted to reduce the spatial correlation of pixels in the stgeo image, and improve the learning mode from supervised learning to self-supervised learning. Secondly, centrally masked convolutions and dilated convolution residual blocks are merged to eliminate secret messages and avoid image quality degradation. Finally, a refinement module is employed to improve image texture details and boundaries.

Results

A series of experiments show that SS-Net from BOSSbase test sets is able to balance the destruction of secret messages with image quality, achieving 100% blocking rate of stego image. Meanwhile, our method outperforms the state-of-the-art methods in secret messages elimination ability and image quality preserving ability.

Introduction

Information hiding is the process of concealing secret messages in digital cover media, so as to achieve the purpose of the transmission of the secret messages in it. As the main implementation of information hiding in covert communication, image steganography (Çiftçi & Sümer, 2022; Kordov & Zhelezov, 2021; Nezami et al., 2022) is the embedding of secret messages in cover image, making the stego image perceptually imperceptible and statistically undetectable. As the antithesis of steganography, steganalysis (Ayaluri et al., 2021; Tabares-Soto et al., 2021a; Tabares-Soto et al., 2021b) aims to reveal the presence of secret messages in suspicious images. Steganalysis can be divided into three levels. The first level goal is to judge whether a suspicious image hides secret messages or not, which is a binary decision problem (Ma et al., 2023a; Ma et al., 2023b). The second level objective of steganalysis is to extract secret messages in stego image. The above two layers of objectives are called passive steganalysis and the image is not modified in any way. The third level is to disrupt or erase secret messages, also known as active steganalysis or steganography sterilization (Wei et al., 2022). Sterilization of steganography attempts to weaken the strength of the secret messages or to sabotage the secret messages in stego image without causing degradation of image quality, making it possible to achieve the goal of successfully blocking covert communication, which is an important practical application in the field of security.

Existing steganography sterilization methods can be divided into two categories: traditional steganography sterilization methods and deep learning based-steganography sterilization methods. Traditional steganography sterilization methods refer to the elimination of secret messages by changing pixel values of an image using various pixel distribution rules. A destroying steganography method was proposed by Ganguly, Mukherjee & Pati (2023), and it can adaptively modify the pixel values of the image based on Unnormalised Hellinger Distance (UHD) and Integer Wavelet Transform (IWT) to eliminate secret messages. This method reduce unnecessary image degradation by selectively modifying pixel values. However, pixel-level logical operations lead to their increased computational complexity at the same time. To avoid the increase of computational complexity, Ameen & Al-Badrany (2021) chose to use filtering techniques and discrete wavelet principles to process the waveform of the image, effectively removing the secret information in the image. Although the quality of the filtered image is improved, the damage to the image content characteristics during the processing is irreversible. In view of this defect, Geetha et al. (2021) found that the diffusion process of steganography information can be effectively destroyed in the image Fourier domain, eliminating most steganography information while maintaining high-quality image content. The above methods use the principle of machine learning to continuously improve learning and overcome the defects in the process of steganography sterilization. However, due to the enhancement of concealment of steganography algorithm, the development of traditional steganography sterilization research (Durdu, 2021) is gradually slowing down.

Steganography sterilization method based on deep learning usually use the powerful representation learning capability of the network to perform a series of active defence tasks to filter and overwrite the secret messages embedded in the image so as to eliminate the secret messages. A generic robust steganography disruption method based on deep learning network (AO-net) was designed by Zhu et al. (2021), which an attack module and an optimization module were included. In order to balance the above two modules, a new loss function is proposed to achieve the purpose of removing secret messages. AO-net can effectively remove secret messages embedded in four robust steganography algorithms including Dither modulation based adaptive steganography (DMAS) (Zhang et al., 2018). Subsequently, A general framework for image secret messages elimination was also proposed by Zhu et al. (2023) in online social networks, which contains two deep learning networks: Scaling-Net and secret clean network (SC-Net), where Scaling-Net is suitable for oversized images in social networks and SC-Net is suitable for normal-sized images. This method can effectively remove secret messages embedded by robust steganography and classical watermarks, and obtain good quality images. However, the model complexity of steganography sterilization methods is high and requires a large amount of arithmetic power with known pairs of cover-stego training data, resulting in a high time-cost overhead.

Considering the above researches, we propose an effective sterilization of image steganography method using self-supervised (Makarov et al., 2022) convolutional neural network (SS-Net). SS-Net is able to reduce the computational overhead and storage overhead while achieving the goal of improving the network training efficiency and reducing the reliance on cover-stego image pairs. Without the need of domain knowledge, SS-Net able to eliminate secret messages quickly while avoiding the degradation of image visual quality. The contributions of this paper are as follows:

• In view of the fact that stego image cannot be directly used as the input and target values for self-supervised training, this paper proposes a purification module and a refinement module to improve the learning strategy of network architecture, thus extending the application scope of the network from supervised learning to self-supervised learning. It enables the model to be trained without ground truth data of cover image.

• Since ordinary convolution can learn potential secret messages, it leads to the mapping of secret messages to the network output layer, which affects blocking effect. Therefore, we adopt the pixel-shuffle down-sampling in purification module, avoiding the learning identity by decomposing the spatial correlation of stego image. In particular, we fuse centrally masked convolutions and dilated convolution residual blocks in purification module, aiming to improve the removal effect of secret messages.

• Considering the visual artefacts in the local details of image generated by purification module. We design refinement module that aims to eliminate secret messages while including the improvement of image texture details and boundaries, smoothing flat regions and avoiding the generation of artefacts. This module is capable of secondary sterilization of stego image with low visual degradation to improve the blocking rate of this method in practical applications.

The organization of this article is structured in the following order: the proposed method (SS-Net), the experimental results and analysis, conclusion and future work.

Materials & Methods

Aiming at the limitations of existing steganography sterilization methods mentioned in the introduction, this paper proposes SS-Net based on purification module and refinement module, which is dedicated to eliminating secret messages of potentially stego images in the social network, and avoiding the degradation of image visual quality to achieve the purpose of active steganalysis. This section is elaborated from three aspects: SS-Net principle, self-supervised learning SS-Net, purification module and refinement module. Firstly, in order to alleviate the phenomenon that the existing active defense methods rely too much on the original carrier image, the learning strategy of the proposed network model is improved from supervised learning to self-supervised learning, which improves the practicability of the method in social network. Secondly, the advantages of the center mask convolution and dilated convolution residual blocks is capture multi-scale image information and extract different dimensional features, so used it to ensure image quality. Finally, the principle of pixel shuffle sampling was used to further eliminate the redundant information in the stego image, to achieve the purpose of eliminating the secret information, so as to realize the active defense and secondary active defense of information hiding.

SS-Net principle

The network models of steganography sterilization methods require paired training data. However, in social networks, it is impossible to capture the stego image corresponding to the cover image, and the network models of these methods cannot be trained when the cover image is not available. Therefore, this paper proposes SS-Net to improve the learning strategy, improve the learning strategy of this model from supervised learning to self-supervised learning, alleviate the phenomenon of the existing steganography sterilization methods that rely excessively on the original cover images, and improve the practicality of this method in practical social networks. The flow of SS-Net is shown in Fig. 1.

From Fig. 1, the SS-Net includes two parts, the first part is purification module, which is able to eliminate the secret messages while preserving the quality of the image. The second part is refinement module, which improves the image generated by purification module, and is capable of secondary filtering of secret messages with low visual degradation to improve the blocking rate of our method in practical applications. Since the refinement module has no training parameters, it does not participate in the network training process. Finally, the sanitized image xsan generated by SS-Net can only extract the wrong secret messages.

Our method aims to eliminate the secret messages without affecting the image quality. The method has two objectives: maintain the visual quality of the sanitized image; and ensure that no valid secret messages can be observed after active defense (i.e., the difference between the secret messages before embedding and the secret messages of the sanitized image extracted after active defense should be large enough). Based on these two objectives, this paper formally defines the sanitized image xsan by Eqs. (1), (2) and (3). (1) xsan=fSS−Netxstego

(2) fSS−Netxstego=freffpurxstego,θ

(3) maxxpurfmesDexstego,Dexsans.t.fpicxstego,xsan≤ɛ

Figure 1 The overall process of SS-Net.

SS-Net includes a purification module and a refinement module.

where xcover and xstego denote the cover image and the corresponding stego image. fSS−Net, fpur and fref denote the overall method of this paper, the first part of this method and the second part of this method, respectively, fmes denotes measure of the difference between secret messages, fpic denotes measure of the visual distance between the two images. ɛ represents an acceptable threshold, and θ denotes a vector of parameters that can be learned by the network model. Equations (1), (2) and (3) is to measure the distance between sanitized and stego images in two domains: message difference and visual difference.

Self-supervised learning SS-Net

Since steganography starts with hiding communication facts, the stego image is highly correlated with the cover image. In this paper, the steganography embedding model and extraction model are defined as En and De. The embedding and extraction process can be represented as Eqs. (5) and (8) respectively. Under the additive model, the stego image distribution after embedding the secret messages is shown in Eqs. (4), (6) and (7). (4) xstego=Enxcover,m

(5) Enxcover,m=xcover+ ∑opei∈Oopei

(6) ϕEnxcover,m=ϕxcover+ ∑opei∈Oδopei

(7) δopei=ϕxcover+opei−ϕxcover

(8) m′=Dexstego

where m and m′ denote the embedded secret messages and the extracted secret messages, respectively. opei∈O,i=1,…,n denotes the basic operation of the secret messages embedding process. O is a combination of operations to complete an steganography algorithm. ϕ⋅ denotes the operation of calculating the statistical distribution. δopei denotes the effect of one embedding basic operation on the distribution of the cover image. If stego images generated by different steganography algorithms is used to train the model with the corresponding cover image, the model will perform poorly on the real stego image under the unknown steganography. Equation (6) describes the cover distribution after steganography, which is an additive model that simplifies the study (Chonev & Ker, 2011). As shown in Eq. (6), each step of operation within secret messages embedding model En has no interactions with each other, and the effects on the cover image distribution are independently additive, such that secret messages are given independent signal for pixels in the cover image, which is able to provide basic assumptions for the training SS-Net. The assumptions are that the secret messages are not related to the cover image. The interpretation of self-supervised learning framework on steganography destruction refers to the use of self-supervised learning methods to detect, identify or restore images or data subjected to steganography attacks. Steganography is a technique for embedding hidden information into overlaying media, such as embedding secret messages into images, audio, or text, without causing a change in human perception. Self-supervised learning frameworks can identify or restore changes caused by these steganographic attacks by learning hidden patterns or anomalies in the data. In self-supervised learning frameworks, a self-supervised task is often designed to simulate the effects of steganography, such as generating labels by adding hidden messages to an image or making changes to the image. The model learns a mapping function that maps the original data to the stego-damaged data to recognize or restore the stego-attacked image. Therefore, this paper proposes a self-supervised approach to train SS-Net, which is a variant of traditional CNN where the center pixel is not visible in the sensory field to predict the corresponding output pixel. We discard the training method of mapping the input of the stego image with secret messages to the clean cover image, and are able to train without the ground truth data of the cover image, which improves the effectiveness of method in social networks.

The self-supervised approach to training SS-Net allows to train directly on the stego image, from which two parts of the training sample, namely the input value and the target value, are derived. In order to clearly demonstrate the use of self-supervised approach to train this network, this paper compares the ordinary network of steganography sterilization with the SS-Net and gives a visualized example as shown in Fig. 2.

Figure 2 Comparison between self-supervised SS-Net and ordinary network training methods.

(A) Ordinary network. (B) SS-Net.

In Fig. 2A, a patch is simply extracted as an input and its center pixel is used as a target, and the prediction of individual pixels relies on a square neighborhood of the input pixel called the pixel’s receptive domain. The model directly maps the values at the center of the input patches to the output values. In Fig. 2B, since SS-Net requires that the center pixel is not visible in the receptive field, SS-Net sets the center of input patch as a blind spot. The only one pixel is removed. Rely on other receptive fields of the input pixels instead of the blind spots, thus avoiding learning consistency. It makes it possible to learn to delete pixel-independent secret messages. SS-Net allows the extraction of input blocks and target values from visual training images. This paper can train it by minimizing the empirical risk by Eq. (9). (9) arg minθfSS−Netxstego,θ−xstego1.

Purification module

Purification module consists of pixel-shuffle down-sampling (Zhou et al., 2019) and purification network based on blind spot network (Krull, Buchholz & Jug, 2018). Pixel-shuffle down-sampling employs the idea of divide-and-conquer and is able to restrict the front-end and back-end of the purification network, making purification network suitable for steganography sterilization. Pixel-shuffle down-sampling rearranges the image pixel values according to the sampling factor β, decomposes the spatial correlation of the image, and is used to isolate the correlation between the image and secret messages, so as to make the pixel points where the secret messages originally exist to be misaligned, and thus removes the high-frequency components that are rich in information, and then eliminates the obvious secret information. The size of β can affect the spatial correlation of the image, and when β is small enough, it can retain the texture structure in the image, but it cannot achieve the purpose of eliminating the secret information. When β is large enough to cause spectral aliasing in the image, the output is a low-frequency signal that does not actually exist, resulting in significant degradation of the image. We set β to 2 in order to balance the two above. The process of pixel-shuffle down-sampling is shown in Fig. 3. Pixel-shuffle down-sampling decomposes the image patch into 2*2 sub-patches, and the four sub-patches are recombined by tiling from Fig. 3.

Figure 3 The process of pixel-shuffle down-sampling.

Blind spot network outputs a result at a point without being able to see the information at that point but only the information at its surrounding points. Purification network is unable to deduce secret messages at the blind spot from the information of the surrounding pixel points, but will only predict the image signals associated with the surrounding structure. Adding the above strategy to a standard CNN network will inevitably hinder efficient training of the network. Therefore, we propose a purification network that integrates centrally masked convolutions and dilated convolution residual blocks to alleviate the above problem. Meanwhile, this paper employs a two-branch blind spot network, which is able to jointly restore more detailed visual information of images through a tightly connected structure. The overall architecture of the Purification module is shown in Fig. 4.

Figure 4 The overall architecture of the purification module.

We utilize the xPD generated by the pixel-shuffle down-sampling strategy as inputs for self-supervised training of the purification network, and feed them one by one into the network to learn the image mapping, which influences each pixel in the output of the network to predict the xpur through the special receptive field of the input pixel. In Fig. 4, firstly, after one layer of convolution, the feature map with the same size as the input image is obtained. Secondly, after Type 1 branch and Type 2 branch respectively, the feature maps obtained from the two branches are spliced by columns without adding new dimensions. The two branches are able to work together to eliminate potential secret messages in the image through a tightly connected structure and dual saliency mechanism, while learning more detailed visual messages and generating efficient feature representations. Finally, convolution layers are passed with the aim of outputting three-channel RGB image. Branch is started by the centrally masked convolution layer. Next, after two sets of convolutional layers. Then, after nine sets of dilated convolution residual blocks and convolution layer. Purification network configuration details are known from Table 1. In Table 1, n represents the number of convolutional kernels, h represents the height of the convolutional kernel, w represents the width of the convolutional kernel, p represents the number of pixels to fill the edges, s represents the stride of the convolutional kernel, and d represents the dilation rate. Centrally masked convolution is introduced to return the corresponding weight of the center pixel to 0, so that the center element is masked on the patch of the original tensor and the connection with the neighboring pixels is restricted, so as to achieve the purpose of eliminating secret messages in the center pixel. The centrally masked convolution is defined as follows: (10) fki=fki−1∗wki−1∘mask

Table 1 Purification network configuration details.

Process	Kernel n*(h*w)	Padding (p*p)	Stride (s*s)	Dilation (d*d)	
Conv	128*(1*1)	(0*0)	(1*1)	(0*0)	
Centrally Masked Conv1	128*(3*3)	(1*1)	(1*1)	(0*0)	
Dilated Conv1	128*(3*3)	(2*2)	(1*1)	(2*2)	
Centrally Masked Conv2	128*(5*5)	(2*2)	(1*1)	(0*0)	
Dilated Conv2	128*(3*3)	(3*3)	(1*1)	(3*3)	

where fki denotes the feature mapping of the k channels of the i-th layer, fki−1 denotes the feature mapping of the k channels of the i-1 th layer, wki−1 denotes the k convolution kernels, ∗ and ∘ denote the convolution operation and the elemental product, respectively, and mask denotes the binary mask which is the same as the size of the convolution kernel.

Dilated convolution residual blocks are able to obtain a larger receptive field, which makes the contact to the image range large, and contains features that do not tend to be localization and detail. Purification network will not learn potential secret messages, and thus avoid mapping the secret messages to the network output layer. At the same time, the dilated convolution residual blocks can capture multi-scale image information, and maintain the output feature map resolution, thus reducing the image quality degradation problem. Dilated convolution residual block is composed of a set of network layers, including the main path and skip connection. The main path consists of dilated convolution layer, convolutional layer and activation function for learning feature representation. The skip connection directly bypasses the main path by adding the input information to the output. This ensures that the input information is more easily propagated to subsequent layers and helps avoid the problem of vanishing gradients. Dilated convolution is able to more aggressively merge spatial information across the input with fewer convolutional layers by expanding the spacing between the kernel points. Dilated convolution with dilation rate of 2 adds zero between the convolution kernel elements to expand the sensory field, and dilated convolution with dilation rate of 3 adds double zero between the convolution kernel elements to expand the sensory field. In purification network, the dilated convolution 1 residual blocks use dilation rate of 2, and the dilated convolution 2 residual blocks use dilation rate of 3.

Refinement module

Purification module is able to eliminate embedded secret messages. However, there are visual artifacts in the local details of the generated images. In order to alleviate the above problems, refinement module aims to eliminate secret messages while including improving the image texture details and boundaries, smoothing the flat regions and avoiding the generation of artifacts. Refinement module is not involved in training, which greatly improves the training time of our method. The process of refinement module is shown in Fig. 5.

Figure 5 The overall process of refinement module.

As can be seen from Fig. 5, xstego generates xpur through the purification module. xpur isused as the input of refinement module, and the output of refinement module is xsan. The refinement module mainly consists of three parts: the first part fills each sub-image pair of xpur with pixel blocks with secret messages respectively to obtain xsep that separates secret messages channel. The second part performs pixel-shuffle down-sampling to obtain xPD, and the purification module is used to eliminate secret messages from each re-filled image xPD to obtain xpur′. The third part averages xpur′ to obtain an image texture detail image xrefineof xstego, and averages xrefine with xpur′ to generate a purified image xsan.

Results

In this section, the implementation details of the realized method are first described and ablation experiments are performed. Then, a series of experiments are conducted to evaluate the effectiveness and sophistication of our method.

Implementation details

We using ALASKA v2 (Cogranne, Giboulot & Bas, 2020a) and BOSSbase 1.01 (Bas, Filler & Pevný, 2011) datasets, where the ALASKA v2 dataset consists of cover images and stego images generated by three steganography algorithms, including J-MiPOD (Cogranne, Giboulot & Bas, 2020b), J-UNIWARD (Holub, Fridrich & Denemark, 2014), and (UERD) (Guo et al., 2015). The images were generated by a steganographic algorithm designed by the ALASKA project team and also included samples of real-world images. The BossBase 1.01 dataset contains 10,000 color images covering a wide variety of topics and content, such as plants, landscapes, architecture, and more. As shown in Table 2. The images are generated by specially designed steganography tools, and in addition to automatic generation, the dataset also contains some real-world images to increase the authenticity and representation of the dataset. The image size is 512 ×512 pixels and the depth is 8 bits.

Table 2 Comparison of different datasets.

Database	Data volume (sheets)	Image type	Image size	Format	
BOSSbase	10000	Grey image	512*512	PGM	
ALASKA v2	80000	Color image	512*512	PGM	

In this paper, 10,000 images are arbitrarily selected from the above dataset. Among them, training 8,000 images, validation 1,000 images and testing 1,000 images. In order to reduce the computational time complexity and achieve efficient training, all images are preprocessed. Preprocessing: first, the 512*512 stego image is cropped into 256*256 small size patches, the overlap between the small size patches is 128, and 9 small size patches will be generated for each image. Then, convert the small size patches form tensor to numpy. Finally, the dimension of feature image (c, h, w) is converted to (h, w, c).

To train an efficient steganography sterilization network, this paper uses PyTorch, trains SS-Net on an Intel(R) Core (TM) i9-10900 CPU @ 2.80 GHz and an NVIDIA GeForce GTX 1660 Ti GPU. Therefore, there is no need to manually define the weights and biases of the network layers. The parameter settings for specific networks are shown in Table 3. In this paper, 100 images are randomly selected from the testing set of two datasets beforehand, and the three most representative steganography algorithms DMAS (Zhang et al., 2018), GMAS (Yu et al., 2020) and LSB are used for steganography.

Table 3 SS-Net parameter settings.

Parameter name	Value	
Image resolution	256 ×256	
Loss function	L1 Norm	
Optimizer	Adam	
Epoch	30	
Learning rate	0.001	
Batch_size	8	

This paper uses the following well-known indicators to assess the degree of elimination of secret information and the impact on image quality. In this paper, the ability to eliminate secret messages is measured using the Bit Error ratio (BER) (Al-Sultan, Ameen & Abduallah, 2019), which is the ratio of the number of erroneous bits to the total number of bits. A larger BER result indicates that more secret messages are eliminated after steganography sterilization. It is generally accepted that hidden secret messages are eliminated when the BER is greater than 0.2. In this paper, Peak Signal-to-Noise Ratio (PSNR) (Huynh-Thu & Ghanbari, 2008) and Structural Similarity Index (SSIM) (Zhou et al., 2004) are used to evaluate the quality of the image after steganography sterilization. PSNR is an objective metric to assess the distortion of the image. SSIM is an objective metric to compare the objective metrics for comparing the similarity of the quality of the image after active defense steganography sterilization with that of the original cover image. The larger the results of PSNR and SSIM, the better the quality of the image. It is worth noting that the value of each objective metric in the experiment is the average result on the testing set.

Ablation experiment

In order to verify the effect of the depth of this network architecture on steganography sterilization, the number of dilated convolution residual blocks in this paper is selected as 5, 7, 9, and 11, respectively, and 20 DMAS stego images are randomly selected from the testing set, with a quality factor of 95 for the cover images and payload of 0.01, and BER, PSNR, and SSIM of the purified images after this method are computed, which the BER, PSNR and SSIM are calculated to verify the effect of eliminating secret messages and image quality. The experimental results are shown in Table 4.

Table 4 Comparison of results for different dilated convolution residual blocks.

Number of dilated convolution residual blocks	BER	PSNR	SSIM	
5	0.485	42.4335	0.9863	
7	0.4828	42.6633	0.9887	
9	0.499	43.0848	0.9898	
11	0.4957	42.6637	0.987	
Notes.

The effect is better when the number of dilated convolution residual blocks is 9.

As can be seen from Table 4, when the number of dilated convolution residual blocks is 9, the BER value of the sanitized image is 0.499, the PSNR value is 43.0848, and the SSIM value is 0.9898. All the above data show that the optimal effect of steganography sterilization can be achieved when the number of dilated convolution residual blocks is selected as 9. Meanwhile, the BER, PSNR and SSIM values of the sanitized image are closer under different number of dilated convolution residual blocks, indicating that the present network architecture has a greater advantage in steganography sterilization itself.

Objective evaluation

In this section, experiments are conducted for steganography algorithms DMAS, GMAS and LSB to verify the effectiveness of our method. The BER of the sanitized images generated by the SS-Net under different payloads and image quality factors is taken as the mean value for the final experimental results, as shown in Tables 5, 6 and 7.

Table 5 The BER, PSNR and SSIM of the proposed method on DMAS.

Metrics	Quality factor	Payload	
		0.01	0.02	0.03	0.04	0.05	
BER ↑	65	0.4780	0.4815	0.4844	0.4880	0.4828	
75	0.4830	0.4879	0.4782	0.4842	0.4787	
85	0.4798	0.4756	0.4875	0.4873	0.4871	
95	0.4923	0.4975	0.5029	0.5000	0.5025	
PSNR ↑	65	43.559	43.5112	43.4239	43.3711	43.0255	
75	45.4116	45.3758	45.2853	45.2214	45.1398	
85	44.6234	44.6149	44.5879	44.5654	44.5396	
95	45.4346	45.4231	45.5448	45.3710	45.3446	
SSIM ↑	65	0.9878	0.9876	0.9872	0.9869	0.9865	
75	0.9918	0.9916	0.9914	0.9912	0.9909	
85	0.9903	0.9902	0.9901	0.9900	0.9899	
95	0.9918	0.9918	0.9917	0.9916	0.9915	

Table 6 The BER, PSNR and SSIM of the proposed method on GMAS.

Metrics	Quality factor	Payload	
		0.01	0.02	0.03	0.04	0.05	
BER ↑	65	0.5089	0.4997	0.4984	0.5007	0.4976	
75	0.4903	0.4894	0.4944	0.5061	0.4958	
85	0.5030	0.5107	0.4957	0.4958	0.4969	
95	0.4885	0.5009	0.5015	0.4995	0.4986	
PSNR ↑	65	43.5730	43.5086	43.4601	43.3356	43.2174	
75	45.4896	45.4275	45.3617	45.2603	45.2258	
85	44.6475	44.6384	44.6202	44.6030	44.5855	
95	45.3709	45.3217	45.2957	45.2291	45.1237	
SSIM ↑	65	0.9880	0.9879	0.9844	0.9876	0.9874	
75	0.9919	0.9919	0.9918	0.9917	0.9916	
85	0.9903	0.9903	0.9903	0.9903	0.9902	
95	0.9918	0.9846	0.9917	0.9916	0.9914	

Table 7 The BER, PSNR and SSIM of the proposed method on LSB.

Metrics	Quality factor	
	65	75	85	95	
BER ↑	0.35718	0.33748	0.35869	0.34249	
PSNR ↑	48.61129	49.00888	48.99212	48.98105	
SSIM ↑	0.99332	0.99415	0.99415	0.99406	

As can be seen from Table 5, under different image quality factors and payloads, the average BER value of sanitized images is 0.487 after our method actively defends against DMAS, which is 243.5% of the baseline value. The baseline value of BER is 0.2 when the sequence of secret messages can not be recovered. The experimental results show that SS-Net can effectively eliminate secret messages and realize steganography sterilization. As can be seen from Table 6, under different image quality factors and payloads, after the steganography sterilization of GMAS by our method, the BER value of the sanitized images is the smallest of 0.4885, which is 244.25% of the baseline value, when the quality factor is 95 and payload is 0.01bpnac, and this value is far more than the unrecoverable value of secret messages. As can be seen from Table 7, under different image quality factors and payloads, the average BER value of sanitized images is 0.349 after our method actively defends against LSB, which is 174.5% of the baseline value. The above data show that under different steganography methods, this method can achieve the purpose of eliminating secret messages and successfully realizes steganography sterilization.

In summary, the value of BER fluctuates between 0.32 and 0.52 under different quality factors, payloads and steganography algorithms. It is verified that the SS-Net can effectively eliminate secret messages in potentially stego images in social networks and realize the purpose of proactive defense against covert communications without considering the specific steganography algorithm, quality factor, and payload used by the steganographer.

Our conduct a series of experiments for robust steganography to verify the effectiveness of SS-Net in maintaining image quality. This experiment uses PSNR and SSIM to evaluate the sanitized images. Our method actively destroys robust steganography DMAS, GMAS and LSB, and the average values of PSNR and SSIM of sanitized images generated by SS-Net are used as the experimental results, as shown in Tables 5, 6 and 7.

As can be seen from Table 5, the PSNR value of sanitized images on DMAS by this method ranges between 43 and 46, and the SSIM value ranges between 0.98 and 1 under different quality factors with different payloads. It shows that the method of this paper can maintain the quality of the image under different quality factors. From Table 6, it can be seen that on different payloads, the average PSNR and SSIM values of the sanitized images reach 43.419 and 0.9871 at a quality factor of 65. At a quality factor of 75, the average PSNR and SSIM values of the sanitized images were 43.353 and 0.9918. At a quality factor of 85, the average PSNR and SSIM values are 44.6189 and 0.9903. At a quality factor of 95, the average PSNR and SSIM values of sanitized images were 45.2682 and 0.9902. As can be seen from Table 7, the PSNR value of sanitized images on LSB by this method ranges between 48 and 49, and the SSIM value ranges between 0.993 and 1 under different quality factors.

Therefore, the PSNR and SSIM of sanitized images under different steganography algorithms meet the quality requirements of images transmitted by social networks, and all the above data show the effectiveness of the method in this paper to maintain the quality of images.

In addition, the different texture images for the results of our display experiment are shown in Table 8. As can be seen from Table 8, it can be concluded that on different texture images, the BER is above 0.52, the value of PSNR is above 44 and the value of SSIM is above 0.994. It can be seen that this method has achieved good results on both high texture and low texture.

What’s more, we conducted a comparative analysis of the influence of different social networks on steganography destruction in the process of image transmission, as shown in Table 9. In the experiment, we compared the changes of images on five social media platforms: Instagram, Facebook, WeChat, Twitter and YouTube. It can be seen that the images after transmission, BER values are above 0.45, PSNR values are above 42 and SSIM values are above 0.96. It can be seen that this method can meet the requirements of different social media networks.

The experimental effect is shown in Fig. 6. In Fig. 6, the first column is the cover image, the second column is the stego image, and the third column is the sanitized image. It can be seen that there is no change in the visual effect of the image after destruction.

Table 8 Effect of different texture images on LSB algorithm.

	Images with different textures	
Metrics					
BER	0.59701	0.59633	0.55556	0.52825	
PSNR	50.8473	51.3308	44.8439	53.1643	
SSIM	0.9957	0.9974	0.9943	0.9964	

Table 9 Performance effect of SS-Net in five social media platforms.

	Instagram	Facebook	WeChat	Twitter	YouTube	
BER	0.4587	0.5177	0.4941	0.51475	0.5	
PSNR	44.0849	46.0878	46.754	47.2209	42.0059	
SSIM	0.9808	0.9879	0.99	0.991	0.9695	

Figure 6 Illustration of the experiment through the SS-Net method.

Comparative experiment

In order to compare with this method, typical and latest steganography sterilization methods are selected in this section: the AO-Net method (Zhu et al., 2021), DFT method (Geetha et al., 2021) and the SC-Net (Zhu et al., 2023) method. Among them, the AO-Net method is the first generalized method for robust steganography and watermarking sterilization. All the test images used in this section are randomly selected from the testing set and the images are compressed into a frequency domain image with a quality factor of 95 and resized to 512 ×512. Thirty stego images were generated at 0.01−0.05 payloads, respectively. Experiments are conducted to compare this paper’s method using the robust steganography DMAS and GMAS as examples, respectively. It is worth noting that the present method performs self-supervised training only on the stego images, while the AO-Net and SC-Net training requires both the stego images and the corresponding cover images. We present the evaluation results of sanitized images obtained by different steganography algorithms, as shown in Tables 10, 11 and 12. Each quantitative metric has 1 number under it, indicating the result after robust steganography DMAS and GMAS sterilization.

Table 10 The BER, PSNR and SSIM of different method on DMAS.

Methods	Metrics	Payload	
		0.01	0.02	0.03	0.04	0.05	
SC-Net	BER	0.4851	0.4966	0.4979	0.4931	0.4934	
PSNR	31.744	31.726	31.711	31.59	31.679	
SSIM	0.9188	0.9184	0.9178	0.9171	0.9163	
AO-Net	BER	0.4984	0.4853	0.4683	0.457	0.4448	
PSNR	41.689	41.68	41.622	41.579	41.523	
SSIM	0.983	0.9829	0.9824	0.9821	0.9815	
DFT	BER	0.495	0.45	0.455	0.4866	0.5018	
PSNR	45.0925	44.04	42.925	41.8375	41.5125	
SSIM	0.9365	0.944	0.9413	0.9423	0.9275	
Our method	BER	0.45437	0.4845	0.47922	0.47365	0.47611	
PSNR	49.00686	48.9823	48.94952	48.90034	48.85715	
SSIM	0.99421	0.99417	0.9941	0.99401	0.9939	

Table 11 The BER, PSNR and SSIM of different method on GMAS.

Methods	Metrics	Payload	
		0.01	0.02	0.03	0.04	0.05	
SC-Net	BER	0.4997	0.4972	0.499	0.5009	0.4976	
PSNR	31.747	31.735	31.716	31.696	31.676	
SSIM	0.9188	0.9187	0.9185	0.9183	0.918	
AO-Net	BER	0.5002	0.5006	0.4996	0.4957	0.4951	
PSNR	41.724	41.727	41.69	41.701	41.691	
SSIM	0.9832	0.9832	0.983	0.9832	0.9831	
DFT	BER	0.57143	0.52679	0.43114	0.43498	0.51971	
PSNR	44.09	43.0925	41.705	40.57	39.7825	
SSIM	0.8704	0.8856	0.9443	0.9436	0.9346	
Our method	BER	0.470478	0.486261	0.493401	0.480311	0.488106	
PSNR	48.95679	48.91651	48.77832	48.70313	48.60782	
SSIM	0.99423	0.99415	0.99406	0.99389	0.99379	

As can be seen from Table 10, it can be seen that in terms of eliminating secret messages, after using our method to eliminate secret messages embedded in DMAS, the BER of sanitized images is higher than that of DFT when the payload is 0.03, 0.04. In the other payload, the BER values of the sanitized images obtained by our method are all slightly lower than those of the other three methods. In terms of image quality, compared with the AO-Net method, SC-Net method and DFT method, the PSNR value of the sanitized image is increased by 7.1895%, 17.1815% and 5.8240%, and the SSIM value is increased by 1.095%, 7.545% and 10.855%, respectively. It can be seen that compared with other active hidden-removal methods for DMAS, our method can eliminate secret messages with high BER and improve the visual quality of the image.

Table 12 The BER, PSNR and SSIM of different method on LSB.

Method	Metrics	
	BER	PSNR	SSIM	
SC-Net	0.32975	50.34086	0.99615	
AO-Ne	0.47122	45.60663	0.98232	
DFT	0.47114	46.92	0.9428	
Our method	0.4625	49.00888	0.99415	

As can be seen from Table 11, it can be seen that in terms of eliminating secret messages, the BER values of the sanitized images on GMAS by our method are all slightly lower than those of SC-Net, AO-Net and DFT when the payload is 0.01 to 0.05. However, in terms of image quality, compared with AO-Net method, SC-Net method and DFT method, the PSNR of the sanitized image by GMAS is increased by 7.23279%, 17.20979% and 4.86679%, and the SSIM is increased by 1.103%, 7.543% and 12.383%, respectively. It can be seen that our method has a clear advantage over other methods for the active defense of GMAS. It can be seen that our center mask convolution and dilated convolution residual block have obvious advantages in the restoration of image quality compared with the GMAS method, but we still have the disadvantage of slightly lower BER value.

As can be seen from Table 12, it can be seen that in terms of eliminating the secret message, after using our method to eliminate the secret message embedded in LSB, the BER of the sanitized image is higher than that of SC-Net. In terms of image quality, compared with AO-Net method and DFT method, the PSNR value of sanitized image is increased by 3.34258% and 2.08888% respectively. The SSIM values are increased by 1.183% and 5.135%, respectively. It can be seen that compared with other LSB active masking methods, our method can not only improve the visual quality of the image, but also eliminate the secret information with high bit error rate.

Conclusions

Most of the existing steganalysis methods is based on passive defense methods such as steganalytic. They address the problem that detection has too high false alarm rate and miss detection rate at low payload. In the face of unknown steganography, payload and other prior knowledge in social networks, steganalysis cannot effectively block covert communication. Existing sterilization of image steganography methods rely excessively on cover-stego image pairs and have low blocking rate of stego images. We proposed self-supervised learning SS-Net. It can eliminate secret messages without any third party noticing and proactively defend against covert communication in social networks. The key technology routes include purification module and refinement module. The self-supervised learning SS-Net firstly improves the learning method of the network through pixel-shuffle down-sampling strategy, restricts the front-end and back-end of the network, and assists the SS-Net. Secondly, the purification module based on centrally masked convolutions and dilated convolution residual blocks take into account the elimination of the secret messages and avoids the degradation of the image quality. The refinement module can improve the local texture details of the image to further safeguard the image quality while eliminating the secret messages. The experiments verified that the our approach based on self-supervised SS-Net is completely feasible, and can realize the ability to maintain the visual quality of the image while interfering with the normal extraction of secret messages.

At present, the proposed method has shown good results against the traditional steganography algorithms. However, due to the increase of steganography algorithms and carrier types, the proposed method still has great limitations in dealing with steganography algorithms. Therefore, the existing SS-Net network structure needs to be trained and improved, to adapt more steganography algorithms, expand the application scenarios of SS-Net and optimize its generalization ability. At the same time, the active defense capability will be further improved in terms of the accuracy and timeliness of destroying secret messages.

Supplemental Information

Supplemental Information 1 The network branching framework.

Supplemental Information 2 Overall framework of SS-Net.

Supplemental Information 3 Author Addition Justification.

Additional Information and Declarations

Competing Interests

Author Contributions

Data Availability

The authors declare there are no competing interests.

Jinjin Liu analyzed the data, authored or reviewed drafts of the article, and approved the final draft.

Fuyong Xu conceived and designed the experiments, performed the computation work, authored or reviewed drafts of the article, and approved the final draft.

Yingao Zhao conceived and designed the experiments, performed the experiments, prepared figures and/or tables, and approved the final draft.

Xianwei Xin performed the experiments, performed the computation work, prepared figures and/or tables, and approved the final draft.

Keren Liu performed the computation work, authored or reviewed drafts of the article, and approved the final draft.

Yuanyuan Ma analyzed the data, authored or reviewed drafts of the article, and approved the final draft.

The following information was supplied regarding data availability:

The publicly available datasets are available at:

- BOSSBase 1.01: https://dde.binghamton.edu/download. From

Tomas Filler, Binghamton University, USA; Tomas Pevny, Technical University Prague, Czech Republic; Patrick Bas, National Centre for Scientific Research. Lille, France.

- ALASKA: https://alaska.utt.fr

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
