# Peer review of "Sterilization of image steganography using self-supervised convolutional neural network"

_PeerJ Computer Science, doi:10.7717/peerj-cs.2330_

## Round 0.1 · original submission · Major Revisions

Dear authors,

You are advised to critically respond to all comments point by point when preparing a new version of the manuscript and while preparing for the rebuttal letter. Please address all the excellent comments/suggestions provided by the reviewers.

Kind regards,

**Language Note:** The review process has identified that the English language must be improved. PeerJ can provide language editing services - please contact us at [email protected] for pricing (be sure to provide your manuscript number and title). Alternatively, you should make your own arrangements to improve the language quality and provide details in your response letter. – PeerJ Staff
PCoelho

Reviewer 1 ·

Basic reporting

The manuscript presents a novel approach to steganography sterilization, aiming to enhance network security by blocking covert communications within social networks. The proposed method, SS-Net, employs a self-supervised learning framework that does not rely on cover-stego image pairs, addressing the limitations of existing methods. SS-Net consists of a purification module and a refinement module designed to eliminate secret messages without degrading image quality. Through extensive experiments, the authors claim to demonstrat that SS-Net achieves an impressive 100% blocking rate of stego images which would outperforms state-of-the-art methods in both secret message elimination and image quality preservation.
The comparison section needs to be more comprehensive. A broader range of state-of-the-art methods should be included in the comparison to contextualize SS-Net's performance accurately. For example, the authors may consider. Security in Medical Image Management Using Ant Colony Optimization; Steganogram removal using multidirectional diffusion in fourier domain while preserving perceptual image quality; Nested two-layer rgb based reversible image steganography method. Detailed discussions on why SS-Net outperforms these methods would provide valuable insights into its effectiveness.
The manuscript should provide more information on the datasets used for training and testing, including their size, diversity, and source. Details on the training procedure, including hyperparameters and optimization techniques, would enhance the reproducibility of the research.
The manuscript contains several grammatical and typographical errors that detract from its overall readability. A thorough proofreading is necessary to bring the manuscript up to academic publishing standards.

Experimental design

Despite its promising contributions, several areas require critical attention and improvement:
The explanation of the self-supervised learning framework and its implementation in SS-Net could be expanded. Detailed descriptions of the architecture, including layer specifications and activation functions, would provide a clearer understanding of how SS-Net operates and its novelty compared to existing models.

Validity of the findings

While the manuscript claims a 100% blocking rate and superior performance in image quality preservation, the experimental section lacks a rigorous statistical analysis. Including additional metrics such as precision, recall, and F1 score, along with confidence intervals for the reported results, would strengthen the validation.
The ability of SS-Net to generalize across different types of steganography methods and various image qualities should be evaluated to ensure the findings are not limited to specific conditions.

Additional comments

The discussion on the limitations of SS-Net and potential areas for future work is notably absent. Acknowledging the limitations and proposing directions for future research would present a more balanced view of the work and its contributions to the field.
Addressing these areas comprehensively will significantly enhance the manuscript's contribution to the field of network security and steganography.

·

Basic reporting

This papers present a new sterilization method to remove steganography secret messages. The architecture is based on a self-supervised approach. A pixel-shuffle down-sampling is used to elliminate spatial correlation between pixels and then a then a two-branched network is used to restore the visual quality of the images.

The main advantages of the approach are related to the fact that the training is not based on cover-stego pairs, while maintaining the visual quality of the sterilized images.

While the main idea of the paper is interesting and it proved to be effetive, generally speaking, there are some weakenesses, namely:

1) the paper lacks in theoretical validation.
- Equation (6), which is central to the method itself, is not clear and properly explained or proved. \phi is said to be “operation of calculating the statistical distribution”. I can envision several method to do that, some more sophisticsted than others. What is the method used? Is the method linear? Authors should explain this properly.
- Eq. (6) assumes superposition (in terms of a linear system) and this is not proved or provided a citation. So I suggest authors to provide proof of this equation or to cite a refenrence with this demonstration.

2) The set of equations (4)-(8) have other issues:
- The domain O is not explained.
- i=1,L is a list of operations, I suppose. They are not clear. Are they pixelwise? Are they a sum of intenstiy values?

3) In the introduction, the sentence “… to achieve the purpose of confirming the ownership of the cover image” is associated with watermarking, not to steganography.

4) The paper has several repetitions that perturb the reader. There are also many typos and sentences not readable. See section 4.

5) Explanation of equations (1)-(3) does not mention that the main idea of them is to measure the distance between sanitized and stego images in two domains: message difference and visual difference. This is understood from the expressions, but it is not explicitly explained in the text.

6) In the Purification module sub-section, the symbol chosed for the ampling factor is f, which may be confused with the functions used previously. I suggst to change to another symbol.

Experimental design

The experimental design is standard and well performed. However, a few diffetent studies should be considered

7) it is known that steganography methods perform differently in different kinds of images. For instance, the use of a full textured image has completely different results than for a low textured image with large areas of very tiny differences (for instance, landscape with a almost homogeneous ble sky). The image quality perception has very different results for the same steganography method. So I suggest authors to address this topic by experimenting different kinds of images and analyze artifacts in low- and high-textured areas.

8) Social media completely changes the images or videos transmitted usually by applying unknown transformations (not disclosed by social media companies). The main transformation is compression, but others are also applied to the color scheme, for instance. This can hinder sterilization methods, specially when steganography methods are designed to resist to this kind of transformations. I suggest a study to analyze the effectiveness resulting of the proposed method being run with images effectively transmitted through different social media (or by simuating the effects realistically).

Validity of the findings

While the validity is straighforward and follows generally the standard method, there are some issues too:

9) The captions of figures and tables are too laconic. In my opinion caption hould be informative and explain the figures and tables. Only placing explanations on the text does not improve readibility of the text. So I suggest more comprehensive captions.

10) PSNR and SSIM are quantitative measures known to be very poor in terms of measuring real image quality and vsual perception. When compared to human judgment, for instance, they perform very bad in matching the human perception. On the other hand, it is true that there are few, or none, other alternative objective metrics that can match human judgment. So this is not an issue to the paper itself, but to what is missing in the paper. See next comment.

11) The paper does not show example images, which is crucial to validate the method’s effectiveness and mainly because SSIM and PSNR are not goodobjective metrics. I suggest the authors show some examples so that the reader can have a judgment about image quality of sterilized images. This should also include images where the method failed to keep image quality and make some discussion on this topic.

12) Figures 6, 7 and 8 should be tables instead of bar charts. Tales are more readable and informative for this sort of information.

13) Figure 2 is not very informative. To improve it, the caption should explain the main diffrence between a) and b).

Additional comments

There are several typos and unreadable sentencen. Not extensively, I pointd out some:

- typos In lines: 48, 69 (“deep learning-based …”), 144 (it’s missing a comma in the expression), 165 (“train the model”), 187 (“is, only”), 216 (“proposes a …”)247 (“are able to…”), 248 (“range large…”), 304, 355 (“Our paper conduct…”), 411 (remove spare text after Conclusions), Figure 2 (“Prediction”), Figure 7 (“PSNR”)

- unreadable sentences in lines: 152 (so much repetition of words), 189-191, 305-306

- in lines 297-298, I suggest to remove “the most popular deep learning framework in academia, and”. This is not relevant to a scientific paper, although I agree.

---

## Round 0.2 · Minor Revisions

Dear authors,

Thanks a lot for your efforts to improve the manuscript.
Nevertheless, some concerns are still remaining that need to be addressed.
Like before, you are advised to critically respond to the remaining comments point by point when preparing a new version of the manuscript and while preparing for the rebuttal letter.

Kind regards,
PCoelho

Reviewer 1 ·

Basic reporting

The introduction and background section of the manuscript lack depth and fail to provide a comprehensive overview of the current state of research in the field of image steganography and its sterilization. The authors briefly mention the development of steganography technology and its implications for network security but do not sufficiently elaborate on existing sterilization methods.
• Key references and recent advancements in the field are missing, making it difficult to contextualize the proposed method within the broader research landscape. The authors should expand this section to include a more thorough review of the literature, highlighting significant contributions and identifying specific gaps that their work aims to address. Durdu, Nested two-layer rgb based reversible image steganography method; Geetha, Steganogram removal using multidirectional diffusion in fourier domain while preserving perceptual image quality.

Experimental design

• The methods section is arguably the most critical part of any research paper, and in this manuscript, it suffers from several deficiencies: The description of the self-supervised convolutional neural network (SS-Net) is vague and lacks the necessary detail to allow replication. The authors introduce concepts like the "pixel-shuffle down-sampling" and "centrally masked convolutions" without adequately explaining their implementation or providing clear definitions. For instance, the process and rationale behind using pixel-shuffle down-sampling need more clarity. There is a noticeable lack of justification for many of the design choices made in the development of SS-Net. The authors should provide a rationale for why certain network architectures, such as the inclusion of dilated convolution residual blocks, were selected over others. Comparative analysis with alternative architectures should be included to substantiate these choices.
• The experimental setup, including the datasets used (ALASKA v2 and BOSSbase 1.01), is mentioned, but the authors fail to provide sufficient details about the preprocessing steps, training procedures, and parameter settings. Information such as the specific configurations of the convolutional layers, the optimization algorithm used, and the criteria for model evaluation is either missing or inadequately described.

Validity of the findings

• The results section presents several experiments purportedly demonstrating the effectiveness of SS-Net. However, there are several critical issues: The presentation of results is inconsistent, with some tables (e.g., Tables 4, 5, and 6) providing extensive metrics while others are missing crucial details. The authors should ensure that all relevant metrics are consistently reported across all experiments to facilitate a fair comparison.
• While the manuscript mentions that SS-Net outperforms state-of-the-art methods, it does not provide a robust comparative analysis. The authors should include detailed comparisons with other methods, such as AO-Net and SC-Net, using standardized benchmarks and metrics. Visual comparisons of sanitized images should be included to demonstrate qualitative differences.
• The authors present results in terms of BER, PSNR, and SSIM but do not discuss the statistical significance of their findings. Statistical tests should be conducted to determine whether the observed improvements are significant and not due to random chance.

Additional comments

• The discussion section does not focus on the implications of the findings. The authors should critically analyze their results, discussing both the strengths and limitations of SS-Net. Potential reasons for the observed performance should be explored, and the broader impact on the field should be considered.
• The future work section is vague and lacks specificity. The authors should outline concrete steps for future research, including potential improvements to the SS-Net architecture, exploration of additional datasets, and the development of more robust evaluation protocols.

·

Basic reporting

The authors carefully followed the comments of reviewers and changed accordingly the article, while explaining the changes.

Experimental design

The authors carefully followed the comments of reviewers and changed accordingly the article, while explaining the changes.

Validity of the findings

The authors carefully followed the comments of reviewers and changed accordingly the article, while explaining the changes.

Additional comments

As most of the comments where followed and in a positive way, in my opinion the article can be published after minor edits of a few remaining typos.

---

## Round 0.3 · accepted · Accept

Dear authors, we are pleased to verify that you meet the reviewer's valuable feedback to improve your research.

Thank you for considering PeerJ Computer Science and submitting your work.

Reviewer 1 ·

Basic reporting

I am satisfied with the revisions.

Experimental design

I am satisfied with the revisions.

Validity of the findings

I am satisfied with the revisions.

Additional comments

The authors have revised well. The manuscript can be accepted for publication.